# Zooming into the Dark Side of Human Annexin-S100 Complexes: Dynamic Alliance of Flexible Partners

**DOI:** 10.3390/ijms21165879

**Published:** 2020-08-16

**Authors:** Judith Weisz, Vladimir N. Uversky

**Affiliations:** 1Departments of Gynecology and Pathology, Pennsylvania State University College of Medicine, Hershey, PA 17033, USA; jxw7@psu.edu; 2Institute for Biological Instrumentation of the Russian Academy of Sciences, Federal Research Center “Pushchino Scientific Center for Biological Research of the Russian Academy of Sciences”, Pushchino, 142290 Moscow, Russia; 3Department of Molecular Medicine and USF Health Byrd Alzheimer’s Research Institute, Morsani College of Medicine, University of South Florida, Tampa, FL 33612, USA

**Keywords:** annexin, S100 protein, Ca^2+^-binding protein, intrinsically disordered protein, protein–protein interactions, multifunctionality

## Abstract

Annexins and S100 proteins form two large families of Ca^2+^-binding proteins. They are quite different both structurally and functionally, with S100 proteins being small (10–12 kDa) acidic regulatory proteins from the EF-hand superfamily of Ca^2+^-binding proteins, and with annexins being at least three-fold larger (329 ± 12 versus 98 ± 7 residues) and using non-EF-hand-based mechanism for calcium binding. Members of both families have multiple biological roles, being able to bind to a large cohort of partners and possessing a multitude of functions. Furthermore, annexins and S100 proteins can interact with each other in either a Ca^2+^-dependent or Ca^2+^-independent manner, forming functional annexin-S100 complexes. Such functional polymorphism and binding indiscrimination are rather unexpected, since structural information is available for many annexins and S100 proteins, which therefore are considered as ordered proteins that should follow the classical “one protein–one structure–one function” model. On the other hand, the ability to be engaged in a wide range of interactions with multiple, often unrelated, binding partners and possess multiple functions represent characteristic features of intrinsically disordered proteins (IDPs) and intrinsically disordered protein regions (IDPRs); i.e., functional proteins or protein regions lacking unique tertiary structures. The aim of this paper is to provide an overview of the functional roles of human annexins and S100 proteins, and to use the protein intrinsic disorder perspective to explain their exceptional multifunctionality and binding promiscuity.

## 1. Introduction

It is known that members of two large families of Ca^2+^-binding proteins, annexins and S100 proteins, can interact with each other in either a Ca^2+^-dependent or Ca^2+^-independent manner, leading the anexin-S100 complexes possessing biological activities [1]. Despite being Ca^2+^-binding proteins, S100 proteins and annexins are very different, both structurally and functionally, with S100 proteins belonging to a superfamily of EF-hand proteins containing two α-helices linked by a Ca^2+^-binding loop [2], and with annexins containing high affinity Ca^2+^-binding motifs known as annexin type II or type III motifs [3,4]. Members of both families play a number of crucial roles in regulation of different cellular processes, have multiple functions, and can interact with multiple partners, raising an important question of the applicability of classical “one gene–one protein–one function” concept to these proteins and even more important questions on the molecular mechanism of such multifunctionality and exceptional binding promiscuity, which are not too typical for ordered proteins with well-defined 3D structures. 

Intrinsically disordered proteins (IDPs) or intrinsically disordered protein regions (IDPRs) can be defined as functional proteins or protein regions that lack ordered three-dimensional structures [5,6,7,8,9,10,11,12,13,14,15]. These proteins have the ability to bind to multiple partners, which enables them to function in regulation, signaling, and control, where they are commonly engaged in one-to-many and many-to-one interactions [5,7,11,12,13,16,17,18,19,20,21,22]. Disordered proteins or protein regions are often affected by post-translational modifications (PTMs), such as phosphorylation, glycosylation, methylation, and ubiquitylation [23,24], and serve as major targets for the alternative splicing (AS) [25,26,27]. All these means are utilized by nature to control and regulate functions of IDPs or hybrid proteins containing ordered domains and functional IDPRs. However, deregulation of IDPs is dangerous and these structure-less, highly dynamic, promiscuously interacting proteins/regions are implicated in numerous human diseases [28,29,30]. Since multifunctionality and binding promiscuity are typically rooted within the protein intrinsic disorder phenomenon, we looked here at the intrinsic disorder propensities of human annexins and S100 proteins and on the roles of intrinsic disorder in their physiological functions. In this article, available information on structures and functions of human annexins and S100 proteins is briefly considered followed by the description of extraordinary interactability of these proteins. Then, intrinsic disorder predisposition of human annexins and S100 proteins is analyzed. Finally, all these facts are brought together to show how proteoforms and structural polymorphism originating from intrinsic disorder can be used as a clue for understanding the multifunctionality and binding promiscuity of these proteins via the protein structure-function continuum concept.

## 2. Structure and Functions of Human S100 Proteins

Human S100 proteins represent a group of small (10–12 kDa) acidic regulatory metal-binding proteins containing 21 members, S100-A1, S100-A2, S100-A3, S100-A4, S100-A5, S100-A6, S100-A7, S100-A7A, S100-A7-like 2 (or S100-A7B), S100-A8, S100-A9, S100-A10, S100-A11, S100-A12, S100-A13, S100-A14, S100-A16, S100-B, S100-G, S100-P, and S100-Z. This family also includes two long proteins, 904-residue long trichohyalin-like protein 1 (S100-A17, also known as basalin) and 2850-residue-long hornerin (S100-A18), both containing S100 domains in their N-terminal regions. Majority of human S100 proteins are encoded by genes located on the human chromosome 1 (locus 1q21), where they represent a part of the epidermal differentiation complex, a chromosomal region that is frequently rearranged in cancer [31,32] and that contains a cassette of more than fifty genes encoding proteins involved in the terminal differentiation and cornification of keratinocytes. This localization of the majority of S100 genes on chromosome 1 defines their established nomenclature, where the consecutive Arabic numbers placed behind the stem symbols S100-A (e.g., S100-A1), whereas S100 genes located on other chromosomes carry the stem symbols S100 followed by a single letter (e.g., S100-B) [32,33].

Structurally, S100 proteins belong to the EF-hand superfamily [34,35,36,37,38,39,40,41,42], containing two Ca^2+^-binding motifs, the C-terminally located canonical EF-hand with high Ca^2+^-binding affinity and the N-terminal pseudo-EF-hand, also known as S100-specific EF-hand, which is characterized by the presence of two extra residues within the first half of the Ca^2+^-binding loop leading to the changed way of calcium coordination and decreased Ca^2+^-binding affinity [43]. Although amino acid sequences of S100 proteins (and S100 domains of basalin and hornerin) are rather diverse (for human S100 proteins, sequence identity ranges from 12.5% to 94.1%, with mean sequence identity of this protein group being 30.6 ± 9.9%, see Appendix A), these proteins are known to possess remarkable structural similarity. In fact, structures of all the S100 subunits characterized so far represent a four-helix bundle. However, the only monomeric representative of human S100 family is S100G (calbindin-D9k, also known as intestinal vitamin D-dependent calcium-binding protein), whereas the majority of S100 proteins exist as a symmetric homodimer, which is held together by the noncovalent interactions between two helices from each subunit (helices I, IV) that form an anti-parallel X-type four-helix bundle [44] (see Figure 1 showing all known structures of human S100 proteins). It is also known that some of the S100 homodimers are additionally stabilized by the interchain disulfide bridges, thereby forming covalently linked S100 dimers with specific cellular functions [45]. Furthermore, some of the S100 proteins are also able to form noncovalent heterodimers (for example, S100-A1/S100-A4, S100-A8/S100-A9, S100-B/S100-A1, S100-B/S100-A6, S100-B/S100-A11). One should also keep in mind, that the oligomerization potential of S100 proteins is not limited by the formation of homo- and heterodimers, and some S100 proteins are known to form tetramers, hexamers, and octamers that can serve as the active extracellular species needed for the receptor binding [46].

In addition to acting as Ca^2+^ sensors, some S100 proteins can also bind zinc and copper cations [42]. In fact, binding of divalent metal cations plays crucial roles in controlling functional properties of S100 proteins promoting homo- or hetero-oligomerization of S100 proteins and modulating their interactions with various specific partners, e.g., adenyl cyclase, caldesmon, cytoskeletal proteins, glycogen phosphorylase, microtubule associated proteins (MAP), nuclear kinase, and some cell cycle-associated proteins, such as neuromodulin and p53.

Functional diversification of S100 proteins can also be attained by their posttranslational modifications (e.g., phosphorylation [47,48]), binding to unsaturated fatty acids in a Ca^2+^-dependent manner [49], as well as due to their different affinities to bind divalent metal cations (calcium, copper, and zinc), different capabilities to form homo- and hetero-oligomers, specific cell and tissue expression patterns, the ability to localize in the cytosol and nucleus, or being secreted via a specific secretion pathway [50]. Since functionality of S100 proteins was a subject of multiple dedicated reviews [35,36,39,51,52,53,54,55,56], there is no need for the detailed description of the multiple biological activities of these important proteins.

Because of their wide distribution and multifunctionality, S100 proteins contribute to a variety of intra- and extra-cellular processes, which range from Ca^2+^ homeostasis to blood coagulation, cell cycle regulation, cell growth and differentiation, cell migration and/or invasion, cell proliferation, cell survival, motility, organization of cytoskeleton, protein phosphorylation, regulation of enzyme activity, secretion, etc. Therefore, not surprisingly, misbehavior of S100 proteins is linked to a multitude of human diseases, such as allergies, asthma, various cancers, cardiomyopathy, diabetes, inflammatory disorders, neurodegeneration, psoriasis, and rheumatoid arthritis [46,57]. Some of the illustrative examples of the broad involvement of S100 proteins in pathogenesis of various human diseases are given by the overexpression of some S100 proteins in brains of patients with Alzheimer’s disease or Down’s syndrome [58], cancer patients [59], HIV-infected individuals [60], and several rheumatic diseases [61]. All this explains the attractiveness of S100 proteins as potential therapeutic targets [57,62,63,64].

## 3. Structure and Functions of Human Annexins

Annexins represent completely different family of Ca^2+^-binding proteins, which are significantly larger than S100 proteins, able to bind multiple Ca^2+^ ions, and whose Ca^2+^-binding sites are not EF-hand motifs (type I), but specific type II and type III sites Ca^2+^ binding sites located in loops [65]. Annexins constitute an evolutionary conserved multigene family of proteins, which have been described in most eukaryotic organisms [4]. In humans, there are 12 annexins (ANXA1-ANXA11 and ANXA13), of which ANXA7, ANXA11, and ANXA13 are considered the oldest members of the group, whereas the nine descendent annexins (ANXA1, ANXA2, ANXA3, ANXA4, ANXA5, ANXA6, ANXA8, ANXA9, and ANXA10) are assumed to originate from a common ancestor, ANXA11 [4]. Curiously, this evolutionary relationship is evident even from the phylogenetic tree that was built based on the multiple sequence alignment of just human annexins (see Appendix A). Comparison of amino acid sequences of human annexins revealed that they share high sequence identity that ranges from 29.8% to 57.7%, with mean sequence identity of this protein group being 44.3 ± 6.9% (see Appendix A).

A very characteristic feature of all annexins is the presence of several copies of an annexin repeat, which is ~70-residue-long conserved structural element required for calcium and membrane binding function of annexins. The presence of such annexin repeats combined with the capability of a protein to interact with the negatively charged phospholipids in a Ca^2+^-dependent manner constitute major criteria defining annexins as a family. There are four such repeats in all human annexins except for ANXA6 that contains eight annexin repeats [4]. Each annexin repeat represents an easily identifiable element of the internal and inter-annexin sequence homology [3]. Figure 2 shows structural hierarchy of human annexins and represents an NMR solution structure of a single annexin repeat from ANXA1 (Figure 2A), an X-ray crystal structure of the full-length ANXA3 containing four annexin repeats (Figure 2B), an X-ray crystal structure of the monomeric ANXA6 containing eight annexin repeats (Figure 2C), and an X-ray crystal structure of the dimeric ANXA13 (Figure 2D). Figure 2A shows that annexin repeat is folded into a tightly packed structure containing five α-helices. When placed together, four such repeats form a tightly packed and highly α-helical disk with a slight curvature (Figure 2B). The convex side of annexins contains the type II and type III Ca^2+^ binding sites [4,65].

This convex side is involved in the peripheral association of annexins with the phospholipid membranes [4]. The opposite side of this α-helical disk is concave, points away from the membrane and can be utilized for interaction with specific binding partners or homodimerization via the N-terminal regions of annexin protomers (see Figure 2D). Finally, Figure 2C shows that ANXA6, which likely originated from the gene duplication during evolution [66], contains two disk-like shaped core lobes of four repeats connected via a flexible linker [67,68] and able to bind to the phospholipid in both parallel and antiparallel orientation [68]. Because of their high sequence conservation, human annexins are structurally similar. This is illustrated by Figure 3 representing structural alignment of human ANXA2, ANXA3, ANXA4, ANXA5, a C-terminal half of ANXA6, ANXA8, and ANXA13. This figure represents four poses of structurally aligned annexins and shows their remarkable structural similarity (in fact, alignment of these structures over 257 residues is characterized by the RMSD of 1.03Å).

In addition to the conserved repeats forming characteristic disc-like C-terminal core domain that contains Ca^2+^ and membrane binding sites and defines the membrane binding potential, annexins contain N-terminally located head domain characterized by substantial sequence diversity and variable length. In fact, in ANXA1-ANXA11 and ANXA13, this domain includes 41, 32, 17, 13, 14, 19, 184, 20, 40, 16, 199, and 13 residues, respectively. In annexins with short N-terminal domains ranging in length from 13 to 20 residues, this region typically extend along the concave side of the annexin, being involved in hydrophobic interactions with the protein core [4].

These short N-terminal domains represent an important structural and functional unit that unfolds independently of the remaining protein structure [71] and have specific regulatory activities, e.g., affecting the Ca^2+^-dependent phospholipid binding of annexins likely due to the stabilization or destabilization of different conformations of these proteins [4]. Longer N-terminal domains of annexins are involved in Ca^2+^- and phospholipid-dependent interactions with various proteins. For example, the N-terminal residues 10–14 of ANXA1 represent a Ca^2+^-dependent binding site for the S100-A11 protein [72,73]. Similarly, in ANXA2, first 14 residues of the N-terminal domain constitute a binding site for the S100-A10 protein [74].

Annexins were originally described as proteins that bind to and hold together specific biological structures (e.g., membranes) and therefore act as scaffolding or bridging proteins. This property is reflected in the names of this protein family members, since “annexin” term is derived from the ancient Greek ανάάξω that literally means “protein of bringing together.” However, as it is often the case for multifunctional proteins, annexins were described in literature by different names related to their functionality or biochemical properties. Among various names given to annexins are calcimedins (proteins mediating Ca^2+^ signals [75]), calpactins (proteins binding Ca^2+^, phospholipid, and actin [76]), chromobindins (proteins binding to chromaffin granules [77]), lipocortins (steroid-inducible lipase inhibitors [78]), and synexin (for granule aggregating protein [79]), as well as anchorins, calelectrins, calphobindings, endonexins, placental anticoagulant proteins (PAPs), thromboplastin inhibitor, vascular anticoagulant-α (VAC-α) [80], etc.

Overall, it is believed that annexins serve as important constituents of the cellular calcium homeostasis displaying Ca^2+^-dependent structural and functional properties [4]. In line with these considerations, it was shown that annexins (particularly, ANXA1, ANXA2, ANXA4, and ANXA6) might form a sophisticated intracellular [Ca^2+^] sensing system, being able to interact with the plasma membrane as well as with internal membrane systems in a highly coordinated and Ca^2+^-dependent manner, potentially providing means for regulation of other signaling pathways [81]. Being Ca^2+^/phospholipid-binding proteins, annexins have to bind Ca^2+^ first in order to interact with membranes [3]. Although conserved core domains of all annexins can bind phospholipids in a Ca^2+^-dependent manner, there is a remarkable difference between these proteins in their sensitivity to Ca^2+^ and specificity for phospholipid head-groups [3]. Additional level of complexity in membrane binding by annexins is determined by their unique N-terminal domains, which seem to contribute to the specific distribution of these proteins inside the cells [82,83,84]. Furthermore, some annexins e.g., ANXA1, ANXA2, ANXA4, ANXA6, and ANXA7, are not only capable of membrane binding but also mediate membrane vesicle aggregation [3]. Finally, at acidic pH, where the native α-helical structure is destabilized by protonation, some annexins can bind membranes independently of calcium. In this case, instead of peripheral interaction with the membrane, annexins assume a fully membrane-integrated structure with the seven-transmembrane-spanning topology [85,86,87]. Annexins also can function as scaffolding proteins to anchor other proteins to the cell membrane [4].

Because of their ability to bind membranes and proteins, annexins play a number of important roles in different cellular and physiological processes. For example, by providing membrane scaffold, annexins are related to changes in the shape of cells. They also participate in organization and trafficking of vesicles, calcium ion channel formation, cell–cell communication, endocytosis, and exocytosis [88]. Annexins found in the extracellular space can be involved in regulation of apoptosis, coagulation, fibrinolysis, and inflammation [89].

Misbehavior of multifunctional annexins is associated with various diseases. As a result, a special term “annexinopathies” was coined to describe pathological consequences of misbehaving annexins [90]. Several illustrative examples of such annexinopathies are represented below. Deregulation and aberrant posttranslational modifications of a glucocorticoid-regulated ANXA1, which is known to be related to adaptive and innate immunity via participation in the regulation of inflammatory cells and the resolution of inflammation, were linked to autoimmunity (e.g., systemic lupus erythematosus) [91]. Furthermore, abnormal expression of this protein is associated with preeclampsia (which is a pregnancy disease associated with impaired inflammatory response) [92] and is closely related to the occurrence and development of tumors, and metastasis [93]. Since ANXA1 is known to be involved in maintaining blood–brain barrier (BBB) integrity, achieved through co-localization with actin microfilaments present at the tight junctions between cells, deregulation of ANXA1 can be associated with BBB leaking [94] and therefore associated with age-related neurodegeneration [95] and multiple sclerosis [94].

Being widely distributed in the nucleus, cytoplasm, and extracellular surface of various eukaryotic cells, ANXA2 is implicated in various biological processes, such as apoptosis, Ca^2+^-dependent regulation of endocytosis and exocytosis, cell proliferation, focal adhesion dynamics, oxidative stress, interactions between cells and the extracellular matrix, as well as transcription, and translation [96]. ANXA2, likely via its complexation with β_2_-glycoprotein I, is involved in the pathogenesis of an autoimmune disease, antiphospholipid syndrome, characterized by arterial, venous or small-vessel thrombotic events, and recurrent miscarriages or fetal loss [97]. Elevated levels of this protein are also found in other autoimmune diseases and thrombotic associated diseases, such as pre-eclampsia [97]. Increased ANXA2 levels are correlated with invasion and metastasis in a variety of human cancers [98], and deregulation and aberrant expression of this protein are found in a large number of human diseases, such as autoimmune and neurodegenerative disease, antiphospholipid syndrome, inflammation, diabetes mellitus, and a series of cancers [99].

ANXA3 is shown to play a noticeable role in tumor formation, cell proliferation, apoptosis, invasion, metastasis, and drug resistance [100,101,102,103]. ANXA4 is altered in atherosclerotic coronary intima [104]. Curiously, although the levels of ANXA2 are increased in antiphospholipid syndrome [97], ANXA5 is under-expressed in this disease [90]. On the other hand, levels of ANXA5 are elevated in serum of all pregnant women on various stages of pregnancy, in serums of patients with several types of cancer [105], and in blood of patients with chronic disease of kidneys [106]. In the dysfunctional bladder, down-regulation of ANXA6 is related to the decline in the contractile properties of bladder [107]. Reduction in the ANXA7 levels is linked to the aggressive metastatic forms of prostate cancer [108] and is related to the suicidal death of erythrocytes, eryptosis [109]. ANXA8 expression is significantly upregulated in ductal carcinoma in situ (DCIS), which is an early form of breast cancer [110], and this protein was identified as one of the putative biomarkers in breast cancer [111]. ANXA9 and ANXA10 expression is altered in head and neck squamous cell carcinomas (HNSCC) [112]. Mutations and deregulation of ANXA11 are related to the development, chemoresistance, and recurrence of cancers, as well as are found in sarcoidosis and systemic autoimmune diseases [113]. Finally, ANXA13 was identified as one of the best kidney biomarkers for refractory lupus nephritis (LN) [114].

## 4. Interactability of Human Annexins and S100 Proteins

It is clear that multifunctionality and polypathogenicity of human annexins and S100 proteins are related to the ability of these proteins to be engaged in multiple interactions with various partners. In agreement with this statement, Table 1 shows that according to the IntAct database (http://www.ebi.ac.uk/intact/) of binary interactions [115] and the STRING computational platform (Search Tool for the Retrieval of Interacting Genes; http://string-db.org/) [116] generating protein–protein interaction (PPI) networks based on the predicted and experimentally derived information on the interaction partners of a protein of interest, all members of both families of the human Ca^2+^-binding proteins analyzed in this study (annexins and S100 proteins) are involved in multiple interactions. In fact, even according to rather conservative IntAct-based estimations, majority of these proteins (24 of 35) have more than ten binding partners, whereas more relaxed STRING-based evaluation using medium confidence level of 0.4 shows that all human annexins and S100 proteins are expected to have more than ten partners, and four annexins (ANXA1, ANXA2, ANXA5, and ANXA7) and seven S100 proteins (S100A4, S100A7, S100A8, S100A9, S100A12, S100B, and S100A18) have more than 100 binding partners each. To further illustrate this point, Figure 4 represents PPI networks for the most connected annexin (ANXA1, Figure 4A) and most connected S100 proteins (S1007, Figure 4B). These networks were generated by STRING using the highest confidence of 0.9. STRING represents a platform conducting functional enrichment analysis of the protein–protein interaction (PPI) networks and contains PPI-related information for 24,584,628 proteins from 5090 organisms [116]. STRING includes 3,123,056,667 known and predicted interactions, of which 52’857’362 interactions are at the highest confidence with the minimum required interaction score of ≥0.900. These interactions represent direct (physical) and indirect (functional) associations and originate from computational prediction, knowledge transfer between organisms, as well as from the interactions aggregated from other (primary) databases [116]. STRING includes eight types of associations grouped into three classes shown in the corresponding PPI networks by different colors. These three groups include known interactions (derived from curated databases and experimentally determined shown as cyan and pink edges, respectively), predicted interactions (based on gene neighborhood, gene fusion, and gene co-expression indicated as green, red, and blue edges, respectively), and others (interactions derived from text mining, co-expression data, and protein homology shown as yellow, black, and light blue edges, respectively). STRING-generated PPI network as an interactive map containing large quantities of useful information. This STRING-based analysis revealed that the ANXA1-centered PPI network contains 354 nodes connected by 37,162 edges. In this network, the average node degree is 210, and the average local clustering coefficient (which defines how close its neighbors are to being a complete clique; the local clustering coefficient is equal to 1 if every neighbor connected to a given node *N_i_* is also connected to every other node within the neighborhood, and it is equal to 0 if no node that is connected to a given node *N_i_* connects to any other node that is connected to *N_i_*) is 0.942.

Furthermore, since the expected number of interactions among proteins in a similar size set of proteins randomly selected from human proteome is equal to 5810, the inter-BAF PPI network has significantly more interactions than expected, being characterized by a PPI enrichment *p*-value of <10^−16^. Analogous STRING-based analysis of S1007 shows that the corresponding PPI network has 86 nodes connected by 3655 edges. It is characterized by the average node degree of 78.1, the average local clustering coefficient of 1.0, and a PPI enrichment *p*-value of <10^−16^, since the expected number of edges is 180. Similar STRING-generated PPI networks for other annexins and S100 proteins are collected in Appendix A.

Besides being engaged in interaction with large cohort of “external” partners, annexins and S100 proteins are known to interact with each other. This “internal” interactability extends beyond the ability of S100 proteins to for homo- and hetero-dimers and the oligomerization of annexins. In fact, several studies clearly indicated the presence of inter-family interactions and pointed out that the complexes between members of these two families are biologically significant [1,117,118]

Figure 5A illustrates this inter-family interactivity by showing the PPI network between 12 human annexins and 23 human S100 proteins generated using STRING with the medium confidence of 0.4. This network includes 35 nodes and has the following characteristics: number of edges is 173, average node degree is 9.89, averaged local clustering coefficient is 0.51, expected number of edges is 3, and PPI enrichment *p*-value is <10^−16^. Figure 5B shows the distributions of intra- and inter-family interactions for human annexins and S100 proteins. Analysis of this plot indicates that 21 proteins are involved in inter-family interactions. Furthermore, although the majority of S100 proteins (with the noticeable exception for S100A2, S100A10, S100A11, and S100A4) prefer S100 family members, all annexins show clear preference for interaction with S100 proteins. In agreement with the individual STRING profiles characterizing high interactability of different members of the annexin and S100 families Figure 6A represents the global annexin-S100-centered interactome that includes 535 nodes connected by 44,201 edges. This network was generated using STRING platform in the multi-protein mode to analyze the interactions of the 12 human annexins and 23 humans S100 proteins with 500 proteins forming the first shell of the resulting interactome (note that the number of interactors in STRING is limited to 500). In this analysis, the medium confidence level of 0.4 was used. Analysis of this dense PPI network indicated that the resulting interactome is characterized by an average node degree of 165 and shows an average local clustering coefficient of 0.798.

The expected number of interactions for the set of proteins of its size is 6968, indicating this annexin-S100-centered-centered PPI network has significantly more interactions than expected (PPI enrichment *p*-value is < 10^−16^). Figure 6B compares involvement of each member of the human annexin and S100 families in annexin-S100 interfamily network and in a global annexin-S100-centered interactome and shows that there is a weak correlation between these two parameters.

## 5. Intrinsic Disorder as a Common Denominator for Understanding the Multifunctionality of Annexins and S100 Proteins

We are showing here that members of both Ca^2+^-binding families considered in this study possess extensive interactability, forming dense and highly connected PPI networks. Since both families are characterized by considerable structural conservation, such binding promiscuity is rather surprising, since, typically, proteins with similar structures are expected to have similar functions. The solution to this conundrum is coming from considering the protein intrinsic disorder phenomenon. In fact, the ability to be engaged in a wide range of interactions with multiple, often unrelated, binding partners represents a characteristic feature of IDPs and intrinsically disordered protein regions IDPRs; i.e., functional proteins or protein regions lacking unique tertiary structures [5,6,7,8,9,16,18,19,119,120,121,122,123,124,125,126,127,128,129]. As a result of their structural plasticity, these proteins/regions were shown to bind to different targets, adopting different conformations at interaction with distinct targets [9,119,121,129,130]. Therefore, the presence of functional IDPRs represents one of the likely explanations for the binding promiscuity of human annexins and S100 proteins. In line with this hypothesis, some of the regions in various S100 proteins, such as Ca^2+^-binding loops, the linker loop connecting the two sub-domains of the protein (‘hinge’), helix III, and the N- and C-termini, were previously characterized as regions with increased mobility [131,132,133,134]. Similarly, structural flexibility was also described for the N-terminally located head domains of annexins, which are characterized by substantial sequence diversity and variable length, ranging from 20 to 199 residues. As it was already pointed out, although short N-terminal domains are typically involved in hydrophobic interactions with the protein core [4], they represent a unique functional and regulatory unit that unfolds independently of the remaining protein structure [71] and can affect the Ca^2+^-dependent phospholipid binding of annexins via the differential alterations of their different conformations [4]. Longer N-terminal domains were shown to be engaged in Ca^2+^- and phospholipid-dependent interactions of annexins with various proteins [72,73,74], and contribute to the intracellular distribution of annexins [82,83,84]. Curiously, N-terminal regions of some annexins are expelled from the core domain as a result of calcium binding [117].

These observations called for the comprehensive analysis of the intrinsic disorder predisposition of the members of the annexin and S100 families. Such an analysis can be conducted by various predictors of intrinsic disorder, which are the specialized computational tools designed to retrieve the information on the intrinsic disorder predisposition of a query protein from its amino acid sequence alone. The outputs of these tools are disorder profiles showing intrinsic disorder predisposition for each residue in a sequence. A residue/region is classified as intrinsically disordered if it is characterized by the predicted disorder score (PDS) ≥ 0.5, and it is flexible if its 0.15 ≤ PDS < 0.5. The intrinsic disorder status of a query protein is determined based on several measures, such as the overall percent of predicted disordered residues, the number and lengths of its IDPRs, and the mean disorder score.

Figure 7 represents the results of the PONDR^®^ VSL2-based [135,136] analysis of the per-residue intrinsic disorder predisposition of human annexins and S100 proteins. Although the majority of annexins and S100 proteins are characterized by fairly uniform sequence length (the average length of nine annexins is 329 ± 12 residues and 21 S100 proteins contain on average 98 ± 7 residues), there are noticeable exceptions from this rule. Here, ANXA6, ANXA7, and ANXA11 contain 673, 488, and 505 residues, respectively, and S100A17 and S100A18 are 904 and 2850-residue long.

To account for this remarkable length difference between the different family members, we present corresponding disorder-based “family portraits” in two forms, (1) where the disorder profiles of all the family members are overlaid (annexins, Figure 7A) or disorder profiles of longest members are shown (S100 proteins, Figure 7C), and (2) disorder profiles focused on core domains of annexins (Figure 7B) or S100 proteins (**inset** to Figure 7C). Figure 7A shows that long N-terminal head domains of ANXA7 and ANXA11 are expected to be highly disordered, and the C-terminal half of ANXA6 is more disordered than its N-terminal half. On the other hand, peculiarities of disorder distribution within the sequences of core domains of human annexins are fairly conserved, since all of them show remarkably similar disorder profiles (see Figure 7B). This is in sharp contrast with the disorder predispositions of the calcium binding domains of S100 proteins, which, in agreement with previous study [137] are characterized by rather different disorder profiles. On the other hand, Figure 7 shows that on average, all annexins and S100 proteins contain rather high levels of intrinsic disorder.

This is further illustrated by Figure 8 representing 2D disorder plots for these proteins, where the percentages of predicted intrinsically disordered residues evaluated by PONDR^®^ VSL2 [135,136] (PPID_VSL2_) are plotted against the percentages of predicted intrinsically disordered residues evaluated by PONDR^®^ VLXT [138] (PPID_VLXT_). These data can be used to roughly classify proteins as highly ordered, moderately or highly disordered based on their corresponding PPID values. In this accepted PPID-based classification, proteins are considered as highly ordered, moderately disordered, or highly disordered if their PPID < 10%, 10% ≤ PPID < 30%, or PPID ≥ 30%, respectively [139]. Figure 8 shows that none of the annexins or S100 proteins is classified as highly ordered by both predictors (there is no symbols within the 10% square). On the other hand, the majority of annexins (ANXA2, ANXA3, ANXA4, ANXA5, ANXA6, ANXA8, ANXA9, and ANXA10) and three S100 proteins (S100A3, S100A10, and S100A12) are predicted to be moderately disordered by both predictors, and remaining members of these two families are predicted as highly disordered by at least one of the predictors. The fact that all annexins are expected to be at least moderately disordered is rather unexpected, since many of these proteins were shown to have unique structures even in their calcium-free states. Furthermore, Figure 7B shows that disorder propensity is unevenly distributed within the individual annexin repeats, with the N-terminal repeats 1 and 2 being typically more disordered than the C-terminal repeats 3 and 4.

It is likely that the IDPRs found in annexins can be of functional importance. It was already pointed out that the head domains of annexins (note that all of them are predicted to contain high levels of disorder) are known to play crucial biological functions. For example N-terminal regions of some annexins, such as ANXA1 and ANXA2 are expelled from the core domain on calcium binding [117]. Importantly, the N-terminal residues 10–14 of ANXA1 represent a Ca^2+^-dependent binding site for the S100-A11 protein [72,73]. Furthermore, it was shown that the ANXA1 N-terminal peptide (residues 1-26) possesses random coil structure in aqueous solution but fold into an α-helical structure upon binding to the small unilamellar vesicles, suggesting that this N-terminal domain of ANXA1 can serve as a secondary membrane binding site in the process of membrane aggregation by providing a peripheral membrane anchor [140,141]. Similarly, in ANXA2, first 14 residues of the N-terminal domain constitute a binding site for the S100-A10 protein [74]. The head domain of ANXA13 can contribute to the alternative fold of the N-terminal region of this protein, where the first two α-helices and the associated helix-loop-helix motif are converted into a continuous α-helix and are used as to form a domain-swapped dimeric form of this protein [142]. To gain more information on potential functional roles of IDPRs in human annexins, these proteins were subjected to the complementary disorder analysis using the D^2^P^2^ platform (http://d2p2.pro/) [143], which is a database of predicted disorder for a large library of proteins from completely sequenced genomes [143]. D^2^P^2^ database uses outputs of several per-residue disorder predictors, such as IUPred [144], PONDR^®^ VLXT [138], PrDOS [145], PONDR^®^ VSL2 [135,136], PV2 [143], and ESpritz [146]. The database is further supplemented by the data on the locations of predicted SCOP domains, conserved Pfam domains, as well as sites of various posttranslational modifications and predicted disorder-based protein binding sites, known as molecular recognition features, MoRFs [143]. It is known that many disorder-based binding regions are characterized by the presence of less disordered sub-regions, which are not capable of folding on their own, but can undergo binding-induced folding at interaction with its binding protein partner. In disorder profiles, such regions are typically manifested as local “dips” within the regions with high disorder score [147,148]. In D^2^P^2^, the presence of MoRFs is evaluated by ANCHOR algorithm [149,150].

Results of this analysis for 11 human annexins are shown in Figure 9 (D^2^P^2^ does not contain information on human ANXA4). Analysis of these functional disorder profiles reveals that in addition to possessing variable levels of intrinsic disorder, all human annexins contain numerous sites of PTMs and six annexins (ANXA1, ANXA2, ANXA6, ANXA7, ANXA10, and ANXA13) are predicted to have MoRFs. Curiously, positions of MoRFs in ANXA1, ANXA2, and ANXA13 coincide with the positions of the aforementioned functionally important parts of N-terminal head domains of these proteins. Since head domains are longest and most disordered ANXA7 and ANXA10, it is not surprising that each of these proteins have multiple long MoRFs. Analogous conclusions can be made based on the D^2^P^2^ analysis of 22 human S100 proteins (there is no D^2^P^2^ profile for S100A2). Figure 10 shows that almost all of S100 proteins are intensively decorated by various PTMs, and nine of them have MoRFs, with very significant parts of highly disordered S100A17 and S100A18 representing disorder-based binding regions. In addition to various PTMs, sequence diversity of human annexins is further enhanced by alternative splicing (AS). ANXA1 has one experimentally validated isoform (346 residues) and two computationally mapped potential isoforms (205 and 114 residues). Canonical isoform of ANXA2 contains 339 residues, whereas AS-generated isoform is extended to 357 residues due to the M_1_ → MGRQLAGCGDAGKKASFKM that adds a disordered stretch to the N-terminal head domain. Although experimental evidence is available only for one ANXA3 isoform (323 residues), at least five potential isoforms with the lengths of 55, 104, 134, 154, and 284 residues are computationally mapped.

Similarly, ANAX5 has a validated isoform of 320 residues and five computationally mapped isoforms with 35, 163, 220, and 260 residues. In addition to the canonical isoform of 319 residues, ANXA4 can exist as an AS-generated isoform (237 residues) with missing residues 1-82. ANXA6 has two described isoforms (canonical isoform with 673 residues and AS isoform with missing residues 1-32) and nine potential isoforms that are computationally mapped (91, 95, 110, 2 × 129, 150, 156, 330, and 460 residues). ANAX7 has a canonical (488 residues) isoform, an AS-generated isoform that misses residues 146-167, and a computationally mapped isoform with 144 residues. For ANAX8, there are three described isoform, a canonical isoform containing 327 residues and two AS isoforms, where isoform 2 (141 residues) has a DYGS → GQQG in region 138-141 and is also missing residues 142-327, whereas residues 8-69 are missing in isoform 3 (265 residues). There are also three computationally mapped isoforms of this protein that contain 21, 276, and 365 residues. Canonical form of ANAX11 contains 505 residues, its AS-generated isoform (472 residues) is missing a part of the N-terminal head domain (residues 1-33), and two computationally mapped isoforms with 148 and 150 residues. The AS-generated isoform of ANXA13 (357 residues) is different from the canonical isoform (316 residues) by the presence H → HSQSYTLSEGSQQLPKGDSQPSTVVQPLSHPSRNGEPEAPQP substitution that which is expected to be highly disordered. Finally, no isoforms were described for ANXA9 and ANXA10.

Very similar situation is observed for human S100 proteins, some of which also have multiple computationally mapped isoforms. The corresponding information is shown below in a form, where a protein name is followed by the series of numbers corresponding to the length of its isoforms, with number of residues in described isoform shown in bold: S100A1 (**94**, 34, 53, 147), S100A2 (**98**, 64, 95), S100A6 (**99**, 85), and S100B (**92**, 94). For S100A5 two isoforms are produced by alternative splicing, canonical isoform 1 (92 residues) and isoform 2 (110 residues) with M_1_ → MPAAWILWAHSHSELHTVM substitution. No isoforms were described for the remaining S100 proteins (S100A3, S100A4, S100A7, S100A7A, S100A7B/S100A7L2, S100A7, S100A9, S100A10, S100A11, S100A12, S100A13, S100A14, S100A14, S100G, S100P, S100Z, and S100A17/Basalin, and S100A18/Hornerin).

## 6. Intrinsic Disorder and S100-Annexin Aomplexes

Structural and functional aspects of various S100-annexin complexes are covered in excellent dedicated reviews [1,117]. In fact, S100A1, S100A4, S100A6, S100A10, S100A11, S100A12, and S100B can interact with annexins, and ANXA1, ANXA2, ANXA5, ANXA6, and ANXA11 can interact with S100 proteins.

With the exception for S100A12-ANXA5 complex representing a unique pair, generally both S100 proteins and annexins show remarkable cross-reactivity, where, for example, S100A6 forms functional complexes with ANXA2, ANXA6, and ANXA11, whereas S100A11 interacts with ANXA1, ANXA2, and ANXA6. On the other hand, ANXA2 is found in complexes with S100A4, S100A6, S100A10, and S100A11, whereas ANXA6 is shown to interact with S100A1, S100A6, S100A11, and S100B [1]. Such S100-annexin complexes can be formed on either a Ca^2+^-dependent or Ca^2+^-independent manner and have a multitude of functional roles, being involved in the differentiation of gonad cells, regulation of the organization of membranes and vesicles, appropriate disposition of membrane-associated proteins, such as ion channels and/or receptors, and are related to neurological disorders [1]. Importantly, there is a broad system of S100–annexin complexes. Some of these complexes are briefly considered below.

The S100A1–ANXA6 and S100B–ANXA6 complexes are found in the membranes of the sarcoplasmic reticulum, the sarcolemma, and transverse tubules in avian skeletal muscle cell, where they can be involved in the regulation of Ca^2+^ fluxes in skeletal muscle cells [151]. Formation of these complexes takes place at high Ca^2+^ concentrations [152] and is driven by the C-terminal half of ANXA6 and does not involve the C-terminal extension of either S100 protein [152,153].

S100A4 is a metastasis-associated protein that forms a complex with ANXA2 and formation of this complex induces angiogenesis [154]. Various functions of ANXA2 in membrane aggregation and endo- and exocytosis are regulated by Ca^2+^ binding, interaction with membrane, various PTMs, and interaction of its intrinsically disordered N-terminal domain (NTD, residues 2–33) with different proteins, including S100A4 and S100A10 [154,155]. Although both ANXA2-S100A4 and ANXA2-S100A10 complexes are heterotetramers, their formation is driven by rather different utilization of the intrinsically disordered NTD of ANXA2, where in the ANXA2-S100A10 complex, interaction is driven by residues 2–14 of ANXA2 binding to S100A10, whereas in ANXA2-S100A4 complex, the entire NTD is wrapped around the S100A4 dimer [155]. The S100A10–ANXA2 complex represents a heterotetramer [(S100A10)_2_–(ANXA2)_2_], where (S100A10)_2_ is located at the center of the complex, interconnecting two ANXA2 molecules. This complex can be found in a membrane fraction and is implicated in liposome aggregation in vitro and in endo- and exocytosis in vivo [156]. Formation of this complex is driven by the N-terminal residues (Val3, Ile6, Leu7, Leu10) of S100A10 [157,158]. As it follows from Figure 10K, these residues are a part of the N-terminal IDPR. Since S100A10 is the only S100 family members that is unable to bind to Ca^2+^ because of the mutation within its EF-hand motifs, formation of the S100A10–ANXA2 complex is regulated by the ANXA2 PTMs [74,159,160]. Because of its heterotetrameric nature, [(S100A10)_2_–(ANXA2)_2_] was shown to link two different membranes [161]. However, it also can attain a geometry, where two ANXA2 molecules bind to the same membrane, and (S100A10)_2_ dimer is facing away from the membrane, creating a platform for interaction with other proteins [162]. In addition to membrane, [(S100A10)_2_–(ANXA2)_2_] was shown to interact with multiple binding partners both inside and outside the cell, such as tissue-type plasminogen activator in the extracellular space [163], as well as various membrane proteins, such as potassium channels [164,165], serotonin 5-HT1B receptors [166], sodium channel [167], and transient receptor potential channels [168].

Malignant tumors are characterized by high expression levels of S100A6 and ANXA11, suggesting that these proteins are related to cancer biology and regulation of the cell cycle [169,170]. Formation of the S100A6-ANXA11 complex is Ca^2+^-dependent and involves interaction of the N-terminal region of ANXA11 (residues 49-62) with S100A6 [171]. Figure 9J shows that although the 200 N-terminal residues of ANXA11 are intrinsically disordered, this region contains two MoRFs (residues 39-49 and 143-212), one of which includes the aforementioned ANXA11 region engaged in the S100A6 binding. In addition to its role in cell cycle regulation and cancer development, the S100A6-ANXA11 complex can be engaged in triggering a cascade for sex determination via some cell stage-specific events [172].

The S100A10–ANXA1 complex can be found on the early endosomal membranes [173] and in the cornified envelope preparation of human keratinocytes [174]. Since this complex is formed in a Ca^2+^-dependent manner, it was suggested that it can be related to the regulation of some Ca^2+^-dependent cellular events [72]. Formation of this heterotetrameric complex is driven by interaction of the N-terminal region of ANXA1 with S100A10 [72,73]. Figure 9A demonstrates that the intrinsically disordered N-tail of ANXA1 contains a MoRF (residues 7-12), which potentially serves as binding site for S100A10. This is in a great agreement with the results of a focused study on the molecular mechanisms of the S100A10–ANXA1 complex formation, which showed that the N-terminal residues 1-13 of ANXA1 and the C-terminal residues 91-94 of S100A10 (which, according to Figure 10K is intrinsically disordered) are indispensable for the complex formation [73].

## 7. Conclusions: Structure-Function Continuum of Annexins and S100 Proteins; Proteoforms Originating from Intrinsic Disorder and Structural Polymorphism as a Clue for Understanding the Multifunctionality and Binding Promiscuity

Considered is this article are two large families of human Ca^2+^-binding protein that play crucial roles is a wide spectrum of cellular processes. Despite the fact that annexins and S100 proteins are rather structured, they are engaged in multiple interactions with large sets of unrelated proteins. We show here that multifunctionality and binding promiscuity of these proteins likely have their roots in intrinsic disorder phenomenon. In fact, all proteins in these families have variable levels intrinsic disorder, with some being rather substantial. This is an important observation, since IDPs and hybrid proteins containing ordered domains and IDPRs have found multiple functional applications of intrinsic disorder. This is directly related to the spatiotemporal structural organization of IDPs/IDPRs, which is known to be very complex and heterogeneous. In fact, looking at the structure of functional proteins, one can find there foldons (independent foldable units of a protein), inducible foldons (disordered regions that can fold at least in part due to the interaction with binding partners), morphing inducible foldons (disordered regions that can differently fold at binding to different binding partners), non-foldons (non-foldable protein regions), and semi-foldons (regions that are always in a semi-folded form) [11,21,175]. Furthermore, functionality of many ordered proteins depends on local unfolding, indicating that these proteins contains unfoldons (ordered regions that have to undergo an order-to-disorder transition to become functional) [176]. Obviously, because of such intricate mosaic-like structural “anatomy,” IDPs/IDPRs are expected to have defined their distinctive molecular “physiology,” with differently (dis)ordered structural elements possessing characteristic functions [22]. All this represents a foundation for multifunctionality of proteins and their ability to be involved in interaction with, regulation of, and be controlled by multiple structurally unrelated partners [22].

This multifarious, structurally and functionally heterogeneous organization of IDPs/IDPRs uniquely places them at the core of the structure-function continuum concept, where instead of the classical (but heavily oversimplified) “one gene–one protein–one structure–one function” view, the relationships between protein structure and function are described by the more convoluted “one gene–many proteins–many conformational ensembles–many-functions” model [22,177], which is based on the proteoform concept [178] introduced to explain an important observation that the complexity of a biological systems is mostly determined by its proteome size and not by the genome size [179]. In fact, the number of functionally different proteins is known to dramatically exceed the number of protein-encoding genes (e.g., human genomes is approaching 20,700 genes [180], but the actual number of functionally different proteins is in a range of a few million [181,182,183,184,185]). The increased size of the functional proteome over a corresponding genome is determined by multiple factors, ranging from the allelic variations (mutations) and various pre-translational mechanisms affecting genes (e.g., production of numerous mRNA variants by the alternative splicing and mRNA editing) to numerous chemical changes induced to proteins by various PTMs [181,182,183,184,185]. In fact, since PTMs can affect protein activity, folding, interactions, localization, stability, and turnover, they are crucial constituents of the protein structure-function continuum, where generation of the multiple proteoforms of a given protein by various mechanisms (including PTMs) defines the ability of a protein to have a multitude of structurally and functionally different states, proteoforms [15,22,177,186,187]. As a results, a single gene can efficiently encode for a set of distinct protein molecules, giving rise to the aforementioned proteoform concept [178]. Furthermore, in addition to the means increasing the chemical variability of a polypeptide chain, the protein structural diversity can be further increased by intrinsic disorder and functioning [15,22,177,186]. In fact, it was pointed out that even without AS, PTMs, or mutations, any protein, being a dynamic conformational ensemble, represents a set of basic (or intrinsic, or conformational) proteoforms; i.e., molecules with identical amino acid sequences but with different structures and, potentially, with different functions. Obviously, any mutated, modified, or alternatively spliced form of a protein (i.e., any member of the inducible (or modified) proteoforms) also exists as a structural ensemble and thereby represents a set of conformational proteoforms [15,22,177,186]. Finally, since protein function, interaction with specific partners, or placement inside the extremely crowded cellular environment can affect structural ensembles of basic and induced proteoforms, functionality per se can be considered as a factor generating functioning proteoforms [15,22,177,186]. In other words, because of all these factors, any given protein exists as a set of basic, induced, and functioning proteoforms [15,22,177,186]. It was also pointed out that combination of AS, PTMs, and intrinsic disorder represents an important means forthe promotion of the alternative, context-dependent states of gene regulatory networks, thereby serving as a critical tool for controlling of a broad range of cellular responses, including cell fate specification [188].

Let us consider now how all these concepts are applicable to the multifunctional annexins and S100 proteins. As it was discussed in this article, these proteins have ordered domains and IDPRs, thereby possessing foldons and non-foldons. Some of the IDPRs can fold at interaction with binding partners, thereby serving as an illustration of inducible foldons. Finally, at least for two annexins (ANXA1 and ANXA2) it was shown that calcium binding resulted in extrusion of their N-terminal head domains [117] that potentially can fold at interaction with binding partners (members of S100 family). Therefore, these head domains can be classified as unfoldons (they are expelled as a result of calcium binding and likely to become disordered) or inducible foldons (they partially fold at interaction with their binding partners), or even morphing inducible foldons (they might have different structures within parent annexin and in a complex with a partner). Since calcium binding results of structuration of neighboring regions, Ca^2+^-binding sites of both families might serve as examples of inducible foldons. All this shows that all annexins and S100 proteins can be considered as basic (or intrinsic, or conformational) proteoforms. Since members of these two calcium-binding families have multiple AS-generated isoforms and are heavily decorated by various PTMs, annexins and S100 proteins exist as inducible (or modified) proteoforms. Finally, formation of interfamily annexin-S100 complexes crucial for functionality of these proteins (with functions of annexin-S100 complexes being different from functions of their original components) serves as an illustrative example of functioning proteoforms. In other words, intrinsic disorder-based structure-function continuum based on the consideration of the corresponding basic, induced, and functioning proteoforms provides an important clue for understanding the molecular mechanisms of multifunctionality and binding promiscuity of human annexins and S100 proteins.

## Figures and Tables

**Figure 1 ijms-21-05879-f001:**
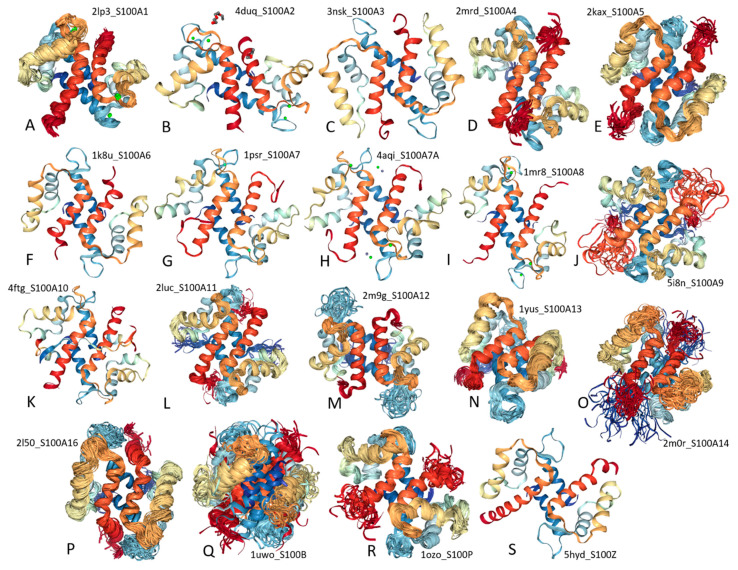
Structural characterization of human S100 proteins. solution NMR structures (plots **A**, **D**, **E**, **J**, **L**, **M**, **N**, **O**, **P**, **Q**, and **R**) or X-ray crystallographic structures (plots **B**, **C**, **F**, **G**, **H**, **I**, **K**, and **S**) are shown for S100-A1 (**A**, PDB ID: 2LP3), S100-A2 (**B**, PDB ID: 4DUQ), S100-A3 (**C**, PDB ID: 3NSK), S100-A4 (**D**, PDB ID: 2MRD), S100-A5 (**E**, PDB ID: 2KAX), S100-A6 (**F**, PDB ID: 1K8U), S100-A7 (**G**, PDB ID: 1PSR), S100-A7A (**H**, PDB ID: 4AQI), S100-A8 (**I**, PDB ID: 1MR8), S100-A9 (**J**, PDB ID: 5I8N), S100-A10 (**K**, PDB ID: 4FTG), S100-A11 (**L**, PDB ID: 2LUC), S100-A12 (**M**, PDB ID: 2M9G), S100-A13 (**N**, PDB ID: 1YUS), S100-A14 (**O**, PDB ID: 2M0R), S100-A16 (**P**, PDB ID: 2L50), S100-B (**Q**, PDB ID: 1UWO), S100-P (**R**, PDB ID: 1OZO), and S100-Z (**S**, PDB ID: 5HYD). Note that no structural information is currently available for S100-A7B, S100-G, basalin (S100-A17), and hornerin (S100-A18).

**Figure 2 ijms-21-05879-f002:**
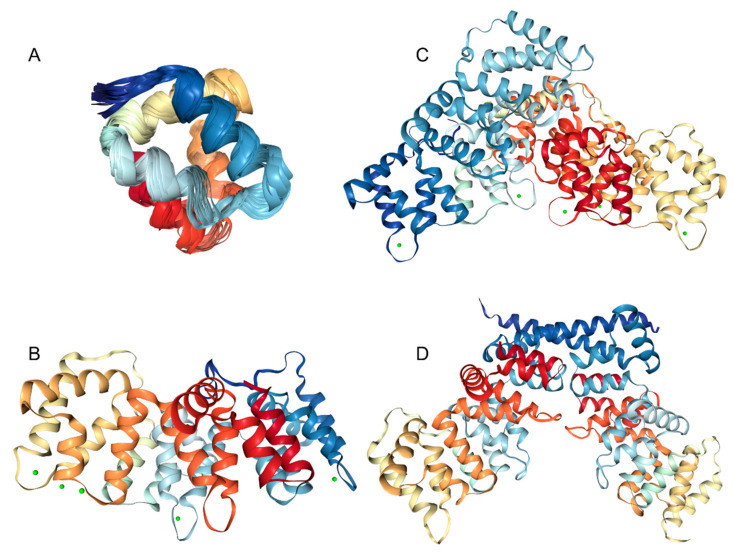
Structural characterization of human annexins. **A**. NMR solution structure of a single annexin repeat from ANXA1 (PDB ID: 1BO9). **B**. An X-ray crystal structure of the full-length ANXA3 containing four annexin repeats (PDB ID: 1AXN). **C**. An X-ray crystal structure of the monomeric ANXA6 containing eight annexin repeats (PDB ID: 1M9I). **D**. An X-ray crystal structure of the dimeric ANXA13 (PDB ID: 6B3I).

**Figure 3 ijms-21-05879-f003:**
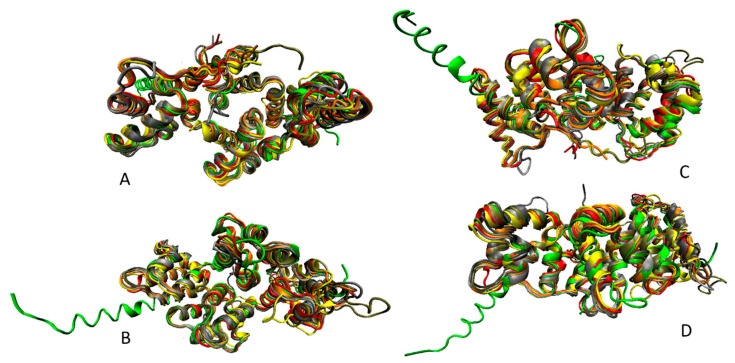
Multiple structural alignment of human annexins (ANXA2, PDB ID: 1W7B, red structure; ANXA3, PDB ID: 1AXN, gray structure; ANXA4, PDB ID: 2ZOC, orange structure; ANXA5, PDB ID: 1AVR, yellow structure; C-terminal half of ANXA6, PDB ID: 1M9I, tan structure; ANXA8, PDB ID: 1W3W, silver structure; and ANXA13, PDB ID: 6B3I, green structure) conducted by MultiProt server [69]. Structures were plotted using the VMD software [70].

**Figure 4 ijms-21-05879-f004:**
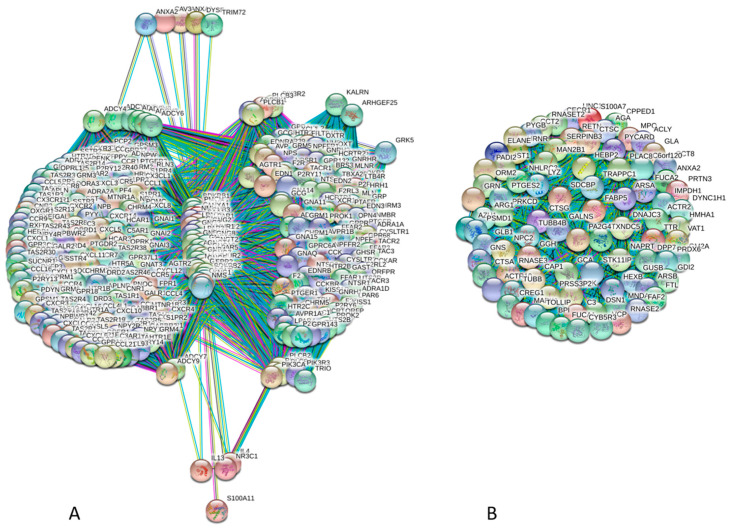
Illustrative examples of highly connected members of annexin and S100 proteins. These PPI networks were generated for ANXA1 (**A**, UniProt ID: P04083) and S1007 (**B**, UniProt ID: P31151) by computational platform STRING using the highest confidence of 0.9. STRING integrates all the information on protein-protein interactions (PPIs), complements it with computational predictions, and returns PPI network showing all possible PPIs of a query protein(s) [114]. The ANXA1-centered PPI network contains 354 nodes connected by 37,162 edges, whereas there are 86 nodes and 3655 edges in the S1007-centered PPI network.

**Figure 5 ijms-21-05879-f005:**
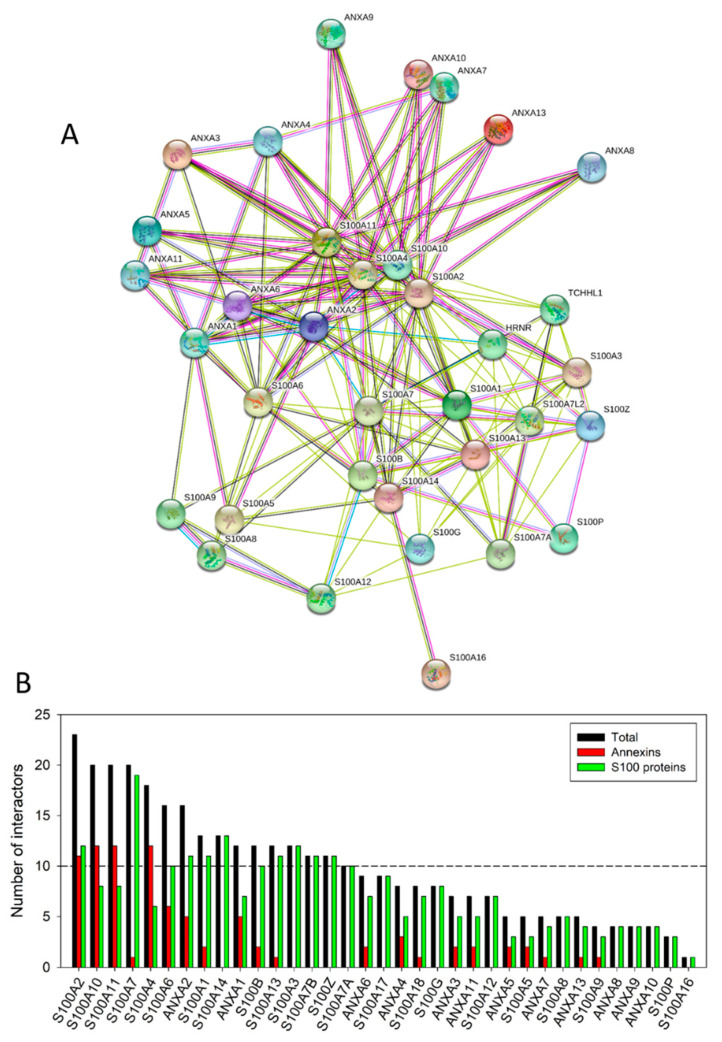
Analysis of the intra-family and inter-family interactivity of human annexins and S100 protein. (**A**) The inter-family interactivity is illustrated by the PPI network between 12 human annexins and 23 human S100 proteins generated by STRING with the medium confidence of 0.4. In this network, 35 human calcium-binding proteins are connected by 173 edges. (**B)** The distributions of intra- and inter-family interactions for human annexins and S100 proteins. For each protein three bars are shown corresponding to the total number of interactions with other annexins and S100 proteins (black bar), the number of its interactions with annexins (red bar) and the number of its interactions with S100 proteins (green bar).

**Figure 6 ijms-21-05879-f006:**
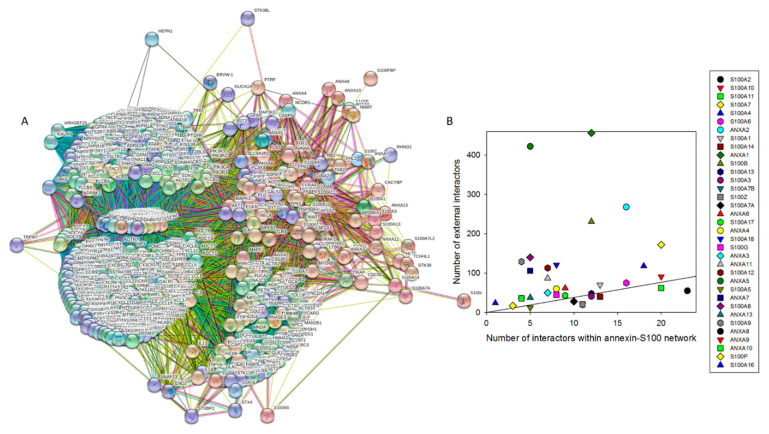
Evaluation of the global interactivity of human annexins and S100 proteins by STRING platform. (**A**) Global annexin-S100-centered interactome that includes 535 nodes connected by 44,201 edges. In this analysis, the medium confidence level of 0.4 was used. (**B**) Comparison of the involvement of each member of the human annexin and S100 families in annexin-S100 interfamily network and in a global annexin-S100-centered interactome.

**Figure 7 ijms-21-05879-f007:**
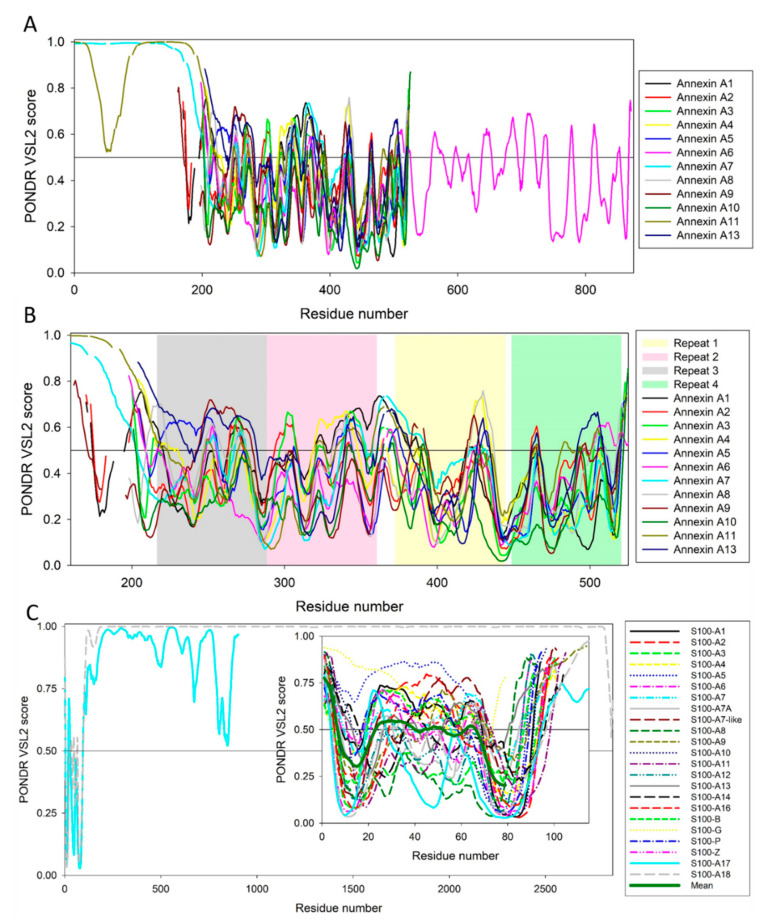
Evaluation of the per-residue intrinsic disorder predispositions of human annexins (**A**,**B**) and S100 proteins (**C**). This analysis was conducted using PONDR^®^ VSL2-based. **Plot A** contains PONDR^®^ VSL2-generated profiles of the aligned annexins. **Plot B** represents disorder profiles of aligned core domains of human annexins. **Plot C** shows disorder profiles of 904-residue long basalin (S100-A17) and 2850-residue-long hornerin (S100-A18). N-terminal regions of both proteins contain short (~100 residues) partially ordered domains, with the remaining 800 and 2750 residues of S100-A17 and S100-A18 being highly disordered. **Inset** to this plot represents PONDR^®^ VSL2 profiles of the N-terminal sections of these two proteins and all other human S100 proteins.

**Figure 8 ijms-21-05879-f008:**
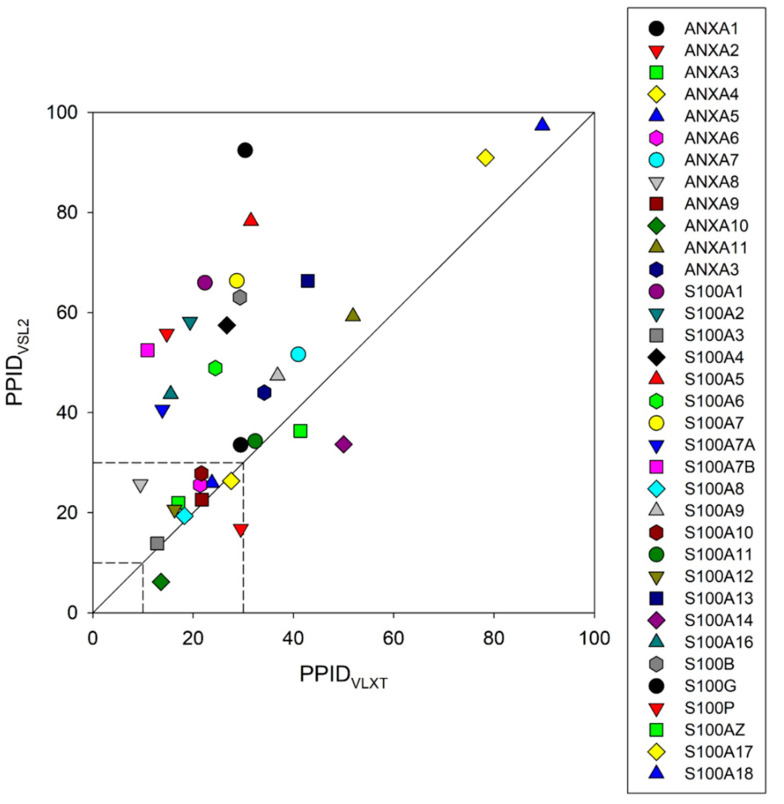
2D representation of the results of evaluation of disorder levels of human annexin and S100 proteins. Here, the percentages of residues in these proteins predicted to be disordered by PONDR^®^ VSL2 are compared with their percentages of disordered residues predicted by PONDR^®^ VLXT. The goal of this plot is to show the overall agreement between the outputs of different disorder predictors used in this study and to show high level of disorder in analyzed proteins.

**Figure 9 ijms-21-05879-f009:**
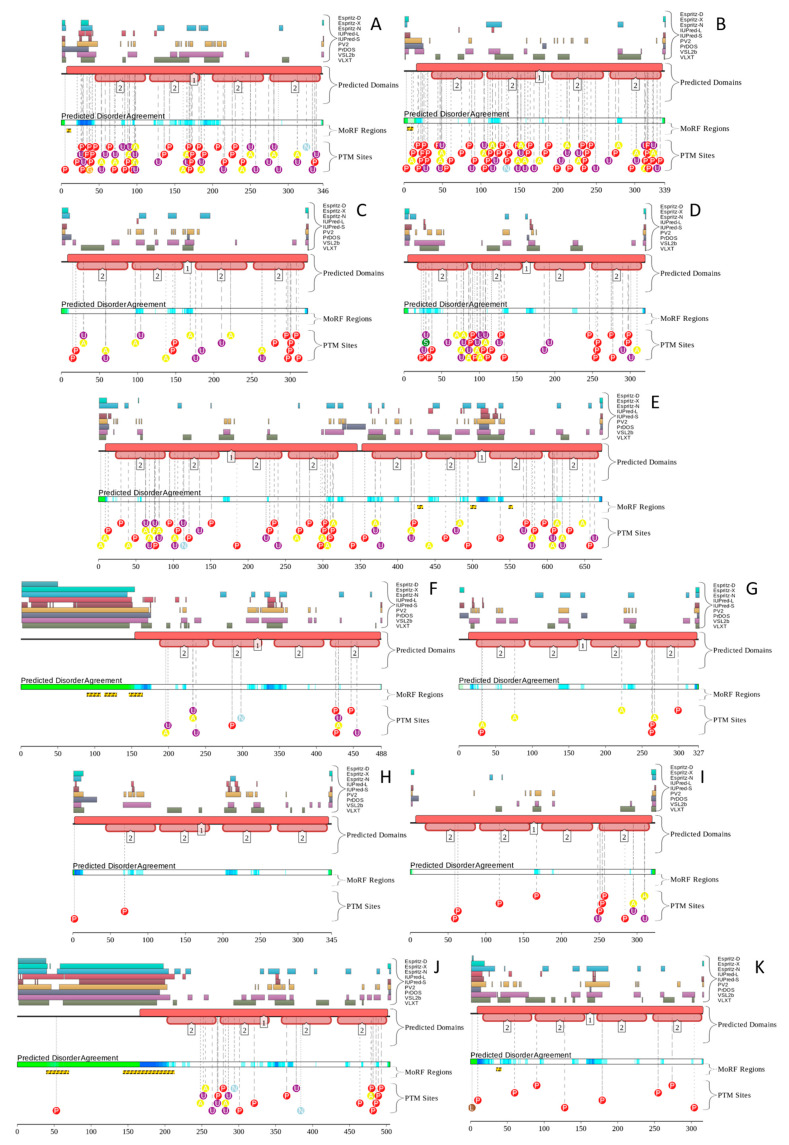
Functional disorder profiles of human annexins generated by the D^2^P^2^ platform (http://d2p2.pro) that generates an interactive display that comments on the structural components, disordered segments, post-translational modifications (PTMs), and the presence of disorder-based binding sites in a protein of interest [139]. **A**, ANXA1; **B**, ANXA2; **C**, ANXA3; **D**, ANXA5; **E**, ANXA6; **F**, ANXA7; **G**, ANXA8; **H**, ANXA9; **I**, ANXA10; **J**, ANXA11; **K**, ANXA13. Note that no D^2^P^2^ profile is currently available for human ANXA4. At the top of each figure, there is a side by side comparison of seven separate disorder predictors (Espritz-D, Espritz-X, Espritz-N, IUPred-L, IUPred-S, PV2, PrDOS, VSL2b, and VLXT), bars indicate positive hits for disorder prediction. Below these colored bars there are two bars showing the position of predicted SCOP domains and conserved Pfam domains. The middle of each plot contains a bar labeled “Predicted Disorder Agreement” presenting the level of agreement between all of the disorder predictors, which is shown as color intensity in an aligned gradient bar below the stack of predictions. The green segments represent disorder that is not found within a predicted SCOP domain. The blue segments are where the disorder predictions intersect the SCOP domain prediction. Below the disorder agreement line, disorder-based binding region (i.e., disordered regions that fold upon interaction with binding partners and known as molecular recognition features, MoRFs) predicted by ANCHOR are displayed as yellow blocks with zigzag infill. Finally, the bottom of the plot shows positions of various PTMs within the query proteins. These are shown by differently colored circles containing letter A (acetylation), L (lipidation), N (nitrosylation), P (phosphorylation), S (sumoylation), and U (ubiquitylation).

**Figure 10 ijms-21-05879-f010:**
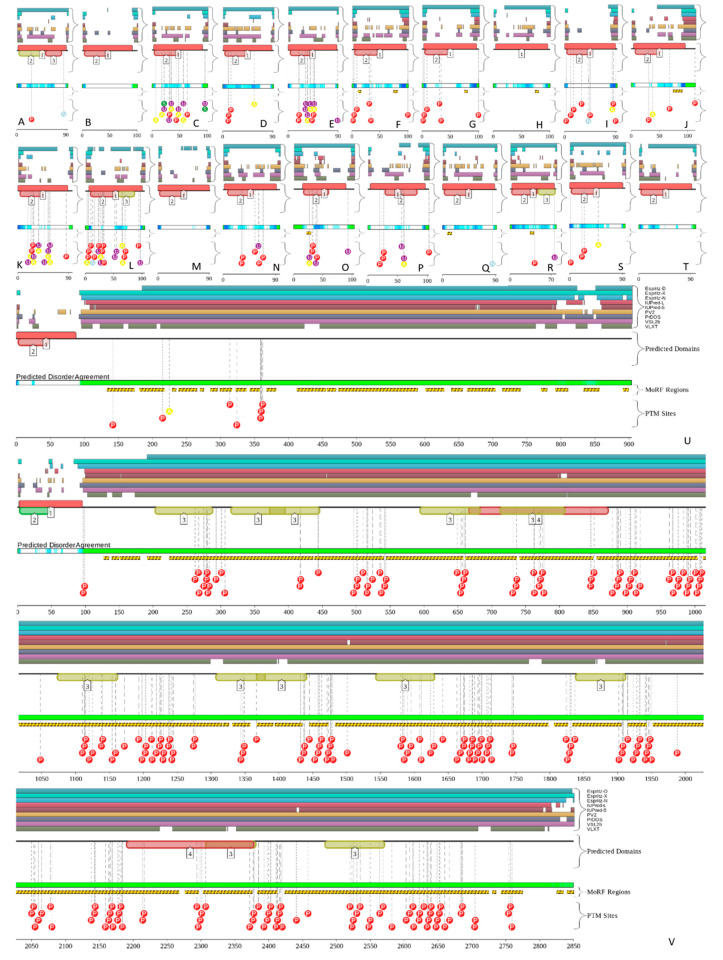
D^2^P^2^-generated functional disorder profiles of human S100 proteins. **A**. S100A1; **B**. S100A3; **C**. S100A4; **D**. S100A5; **E**. S100A6; **F**. S100A7; **G**. S100A7A; **H**. S100A7L2; **I**. S100A8; **J**. S100A9; **K**. S100A10; **L**. S100A11; **M**. S100A12; **N**. S100A13; **O**. S100A14; **P**. S100A16; **Q**. S100B; **R**. S100G; **S**. S100P; **T**. S100Z; **U**. S100A17; **V**. S100A18. All the keys are described in the legend to Figure 9.

**Table 1 ijms-21-05879-t001:** Evaluating interactability of human annexins and S100 proteins.

Protein Name	UniProt ID	N_IntAct_	N_STRING (0.9)_	N_STRING (0.7)_	N_STRING (0.4)_
Annexins					
ANXA1	P04083	89	354	362	456
ANXA2	P07355	102	99	119	268
ANXA3	P12429	7	0	9	50
ANXA4	P09525	11	0	2	60
ANXA5	P08758	52	3	37	422
ANXA6	P08133	24	6	10	62
ANXA7	P20073	122	2	20	106
ANXA8	P13928	13	0	4	36
ANXA9	O76027	5	0	2	38
ANXA10	Q9UJ72	3	0	3	36
ANXA11	P50995	27	0	12	87
ANXA13	P27216	0	0	5	38
S100 proteins					
S100A1	P23297	16	5	21	70
S100A2	P29034	21	0	9	55
S100A3	P33764	20	2	8	41
S100A4	P26447	44	2	13	118
S100A5	P33763	2	0	0	12
S100A6	P06703	27	3	13	75
S100A7	P31151	41	86	94	172
S100A7B	Q86SG5	8	0	3	20
S100A7A	Q5SY68	4	0	0	28
S100A8	P05109	59	8	19	140
S100A9	P06702	52	8	15	129
S100A10	P60903	50	10	20	91
S100A11	P31949	15	3	5	62
S100A12	P80511	2	15	16	113
S100A13	Q99584	18	4	6	48
S100A14	Q9HCY8	15	0	4	40
S100A16	Q96FQ6	12	0	10	24
S100B	P04271	30	19	51	231
S100Z	P29377	0	0	6	21
S100G	P25815	20	0	5	45
S100P	Q8WXG8	8	3	7	17
S100A17	Q5QJ38	0	0	6	43
S100A18	Q86YZ3	32	86	87	121

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
