# Peer review of "Zooming into the Dark Side of Human Annexin-S100 Complexes: Dynamic Alliance of Flexible Partners"

_ijms, 2020, doi:10.3390/ijms21165879_

Round 1

Reviewer 1 Report

Authors have provided a very good review of S100 and Annexin interaction networks. I would suggest authors to dwell more on interaction of annexin with membrane. It would be great if authors can give example of conformational change from disorderness to 7TMH. I would also suggest to dissect different section on processes and diseases for annexins. What physiological information is derived from these string networks. Explain. Currently it is mostly the explanation of network. It would be good to explain the details for preparation of String networks, probably in supplemental. Please explain Intrinsic disorder status in a bit more detail. Figure 9 and 10 are too small to visualize. Improved resolution may be better. 

Author Response

Authors have provided a very good review of S100 and Annexin interaction networks.

RESPONSE: We are thankful to this reviewer for high evaluation of our work.

I would suggest authors to dwell more on interaction of annexin with membrane. It would be great if authors can give example of conformational change from disorderness to 7TMH.

RESPONSE: Thank you for pointing this out. Although the manuscript already contains rather intensive discussion of the interaction of annexins with membranes (see pages 5-8), we added a short statement that ANAX1 can undergo a structural transition from random coil to a-helix in the presence of the small unilamellar vesicles (see page 17):

Furthermore, it was shown that the ANXA1 N-terminal peptide (residues 1-26) possesses random coil structure in aqueous solution but fold into an α-helical structure upon binding to the small unilamellar vesicles, suggesting that this N-terminal domain of ANXA1 can serve as a secondary membrane binding site in the process of membrane aggregation by providing a peripheral membrane anchor [140,141].

I would also suggest to dissect different section on processes and diseases for annexins.

RESPONSE: Thank you for pointing this out. However, we think that physiological and pathological roles of annexins are intertwined and therefore splitting them into the different sections would distort the flow of the text.

What physiological information is derived from these string networks. Explain. Currently it is mostly the explanation of network. It would be good to explain the details for preparation of String networks, probably in supplemental.

RESPONSE: Thank you for pointing this out. Corresponding explanation is added (see page 10):

STRING represents a platform conducting functional enrichment analysis of the protein-protein interaction (PPI) networks and contains PPI-related information for 24,584,628 proteins from 5,090 organisms [116]. STRING includes 3,123,056,667 known and predicted interactions, of which 52'857'362 interactions are at the highest confidence with the minimum required interaction score of ≥0.900. These interactions represent direct (physical) and indirect (functional) associations and originate from computational prediction, knowledge transfer between organisms, as well as from the interactions aggregated from other (primary) databases [116]. STRING includes 8 types of associations grouped into three classes shown in the corresponding PPI networks by different colors. These three groups include known interactions (derived from curated databases and experimentally determined shown as cyan and pink edges, respectively), predicted interactions (based on gene neighborhood, gene fusion, and gene co-expression indicated as green, red, and blue edges, respectively), and others (interactions derived from text mining, co-expression data, and protein homology shown as yellow, black, and light blue edges, respectively). STRING-generated PPI network as an interactive map containing large quantities of useful information.

Please explain Intrinsic disorder status in a bit more detail.

RESPONSE: We are not sure what is requested here. We added the following explanation to page 14:

These observations called for the comprehensive analysis of the intrinsic disorder predisposition of the members of the annexin and S100 families. Such an analysis can be conducted by various predictors of intrinsic disorder, which are the specialized computational tools designed to retrieve the information on the intrinsic disorder predisposition of a query protein from its amino acid sequence alone. The outputs of these tools are disorder profiles showing intrinsic disorder predisposition for each residue in a sequence. A residue/region is classified as intrinsically disordered if it is characterized by the predicted disorder score (PDS) ≥ 0.5, whereas residues/regions with 0.15 ≤ PDS < 0.5 are considered as flexible. The intrinsic disorder status of a query protein is determined based several measures, such as the overall percent of predicted disordered residues, the number and lengths of its IDPRs, and the mean disorder score.

Figure 9 and 10 are too small to visualize. Improved resolution may be better.

RESPONSE:  High resolution figures are submitted

Reviewer 2 Report

This is an interesting paper describing the structurally flexible region of annexins and S-100 proteins.

Although the title says human annexin-S100 complexes, there are very few descriptions on annexin-S100 complexes.  Authors should make a section for annexin-S100 complexes.

Author Response

This is an interesting paper describing the structurally flexible region of annexins and S-100 proteins.

RESPONSE: We are thankful to this reviewer for high evaluation of our work.

Although the title says human annexin-S100 complexes, there are very few descriptions on annexin-S100 complexes.  Authors should make a section for annexin-S100 complexes.

RESPONSE: Thank you for pointing this out. Although annexin-S100 complexes were discussed in several parts of the original manuscript, we added a dedicate section to the manuscript. This section is entitled “Intrinsic disorder and S100-annexin complexes” and is located at the end of the manuscript.